# Tracking of Extracellular Vesicles’ Biodistribution: New Methods and Approaches

**DOI:** 10.3390/ijms231911312

**Published:** 2022-09-25

**Authors:** Alexander M. Aimaletdinov, Marina O. Gomzikova

**Affiliations:** Laboratory of Intercellular Communication, Institute of Fundamental Medicine and Biology, Kazan Federal University, Kazan 420008, Russia

**Keywords:** extracellular vesicles, exosomes, microvesicles, biodistribution, bioluminescence, fluorescence, positron emission tomography, single photon emission computed tomography, computed tomography, magnetic resonance imaging

## Abstract

Extracellular vesicles (EVs) are nanosized lipid bilayer vesicles that are released by almost all cell types. They range in diameter from 30 nm to several micrometres and have the ability to carry biologically active molecules such as proteins, lipids, RNA, and DNA. EVs are natural vectors and play an important role in many physiological and pathological processes. The amount and composition of EVs in human biological fluids serve as biomarkers and are used for diagnosing diseases and monitoring the effectiveness of treatment. EVs are promising for use as therapeutic agents and as natural vectors for drug delivery. However, the successful use of EVs in clinical practice requires an understanding of their biodistribution in an organism. Numerous studies conducted so far on the biodistribution of EVs show that, after intravenous administration, EVs are mostly localized in organs rich in blood vessels and organs associated with the reticuloendothelial system, such as the liver, lungs, spleen, and kidneys. In order to improve resolution, new dyes and labels are being developed and detection methods are being optimized. In this work, we review all available modern methods and approaches used to assess the biodistribution of EVs, as well as discuss their advantages and limitations.

## 1. Introduction

Extracellular vesicles (EVs) are nanosized vesicles released by cells. EVs can be found in various human physiological fluids such as urine, blood, breast milk, saliva, cerebrospinal fluid, amniotic fluid, and synovial fluid [1,2]. EVs are secreted by various cell types, including T cells, B cells, NK cells, dendritic cells, platelets, mast cells, epithelial cells, endothelial cells, neurons, oligodendrocytes, Schwann cells, muscle cells, erythrocytes, cancer cells, embryonic cells, and mesenchymal stem cells (MSCs) [3,4,5,6].

EVs are surrounded by a phospholipid membrane and contain cytoplasmic components of parental cells [3,7]. Extracellular vesicles provide horizontal transfer of microRNAs (miRNAs), mRNA, DNA, proteins, lipids, and other bioactive molecules between cells [3,8]. Organelles such as ribosomes and mitochondria have also been found in EVs [9].

EVs are divided into the following three subgroups according to their biogenesis: exosomes, which represent endosomal vesicles 40–150 nm in diameter that are released into the extracellular environment by the fusion of multivesicular bodies with the cell membrane; microvesicles—vesicles 150–1000 nm in diameter, resulting from direct budding from cytoplasmic membrane; and apoptotic bodies—vesicles releasing as a result of cell death, with a broad size distribution of 50–2000 nm [10,11].

EVs are multifunctional signaling complexes that control fundamental cellular functions [12]. EVs are involved in angiogenesis [13], antigen presentation [14], apoptosis [15], coagulation, cell homeostasis, inflammation [16], cell differentiation, and mediation of intercellular signal transduction [17,18]. EVs not only play an important role in the regulation of normal physiological processes, such as stem cell maintenance, tissue repair, and immune modulation [19,20,21], but also participate in pathological processes, such as cancer, Alzheimer’s disease, Parkinson’s disease, systemic lupus erythematosus, diabetes, rheumatoid arthritis, vitiligo, and pre-eclampsia [22,23,24,25].

EVs are considered as important circulating disease biomarkers [26]. EVs derived from tumor cells are used as effective biomarkers for the type and stage of cancer [27,28]. Furthermore, tumor specific mutations, such as EGFRvIII in glioblastoma [26], and molecular composition (i.e., the presence of specific proteins and nucleic acids) reflecting EVs’ origin are detected [23,29,30,31].

To date, EVs have also demonstrated significant therapeutic potential in numerous disease models [32]. EVs derived from mesenchymal stem cells demonstrated the regenerative potential in the treatment of cardiovascular diseases, kidney, liver, and nervous system injuries, as well as skin wounds [33]. Therefore, the use of cell-free therapies based on EVs is considered a promising approach to stimulate tissue regeneration [34]. EVs overcome the key limitations of cell-based therapies. Their main advantages are stability, low immunogenicity, and dosage flexibility, i.e., the possibility of application in a rather wide range of therapeutic doses [35,36].

Before EVs become clinically approved as carriers for the delivery of drugs and therapeutic agents, their biodistribution and pharmacokinetics need to be carefully studied [12]. Monitoring the biodistribution of exogenously introduced EVs in vivo remains extremely challenging owing to their natural origin, small size, and short half-life [22].

Labelling with lipophilic fluorescent dyes such as PKH26/67, DiD, and DiR [37,38,39,40]; labelling with radioisotopes such as (99 m)Tc-HMPAO) and Indium111 [41,42]; techniques of nuclear and magnetic resonance imaging [43]; as well as genetic engineering techniques to load luminescent proteins such as Renilla, Gaussia luciferases, and firefly luciferase (FLuc) into EVs [44,45,46,47,48] are used to track EVs in vivo.

The aim of this paper is to review recent advances in molecular imaging and methods and approaches to visualize EVs in vivo. We discuss the advantages and disadvantages of the described methods and compare EVs’ biodistribution profiles obtained with different imaging techniques.

## 2. Bioluminescence Imaging

Bioluminescence is widely used in preclinical studies to visualize molecular and cellular processes in normal and pathological conditions. Bioluminescence imaging (BLI) is considered to be the most sensitive method for EVs’ detection in vivo. A major advantage of BLI is a high signal-to-noise ratio (SNR), as mammalian tissue does not normally contain luciferases and does not emit light, i.e., has a negligible auto-luminescence. Therefore, BLI has an excellent signal-to-noise ratio and is an extremely sensitive detection tool [49]. Depending on the type of luciferase, the SNR can reach about 10^2^–10^7^ [50]. It is based on the self-emission of light in the yellow to green wavelength range, as a result of luciferin oxidation catalyzed by luciferase enzyme [51]. To load EVs with luciferase, donor cells are subjected to genetic modification using a reporter construct. The substrate, luciferin, is injected (intraperitoneally or intravenously) into an animal to identify the location of EVs in vivo [52]. 

The most commonly used reporter proteins for BLI are luciferases of North American firefly *Photinus pyralis* (562 nm, green light), the fire nutcracker beetle *Pyrophorus plagiophthalamus* (578 nm, yellow light), the sea pansy *Renilla reniformis* (480 nm, blue light), and the crustacean sea copepod *Gaussia Princeps* (480 nm, blue light). Luciferases from terrestrial organisms use d-luciferin as a substrate and ATP, Mg^2+^, and O_2_ as cofactors, whereas luciferases from marine organisms use coelenterazine analogues as substrates [53]. The new luciferase NanoLuc (460 nm, blue light) is of particular interest. It was isolated from the deep-sea shrimp *Oplophorus gracilirostris* [54]. This bioluminescent platform differs from other known systems by its smaller size and higher luminescence level, approximately 150-fold higher than that of other systems [55].

Gangadaran P. et al. used the BLI method and luciferase isolated from *Renilla reniformis* (Rluc) in their study. The authors investigated the biodistribution of EVs derived from thyroid cancer cells (CAL-62) and breast cancer cells (MDA-MB-231). After intravenous injection, EVs derived from CAL-62/Rluc were detected first in the lungs and then in the liver, spleen, and kidneys. EVs derived from MDA-MB-231/Rluc also showed strong signals in the liver, lungs, spleen, and kidneys. EV-CAL-62/Rluc and EV-MDA-MB-231/Rluc could be detected up to day 3 and day 9 after injection, respectively [56]. 

Kanada M et al. used the bioluminescence technique to investigate the transfer of functional biomolecules through EVs derived from mouse breast cancer cells (MET-1 line) in vivo. EVs were injected intravenously in mice. Bioluminescence signals were observed in the pancreas and the abdominal wall. The signal persisted for 21 days [57].

Despite all of the above-mentioned advantages, the BLI method has a limitation associated with a weak bioluminescence signal obtained from EVs, which is a consequence of their small size. Approaches to improving the bioluminescence technique are currently under development. 

Wu A.Y. et al. used bioluminescence resonance energy transfer (BRET)-based reporters to tag EVs. BRET reporters consist of a BLI protein and a fluorescent protein or molecule that are conjugated in close proximity to each other through a linker. BLI protein catalyzes the oxidation of substrate with concomitant light emission, which is captured by a fluorescent acceptor—that is, intramolecular energy transfer. The intensity of emissions of BLI and fluorescent protein is detected. In the study of Wu A.Y. et al., EVs derived from the 293T cell line were injected intravenously in C3H mice. Images were obtained 5, 10, 20, and 30 min after substrate injection. The authors found that most EVs signals were detected in the lungs and spleen in vivo. Ex vivo imaging revealed signals in the liver (18.29–23.9%), spleen (20.09–12.15%), and lungs (50.66–54.71%), as well as a small signal in the kidneys [58]. 

Takahashi Y et al. used a hybrid protein (gLuc-lactadherin) composed of Gaussia luciferase (as a reporter) and lactadherin (membrane-associated protein found in EVs) to assess the distribution of EVs. EVs derived from murine melanoma B16-BL6 cells (transfected with plasmid encoding the fusion protein) rapidly disseminated from the bloodstream after intravenous injection (half-life of about 2 min) and were detected in the liver and in the lungs [48]. This scientific group further detected the bioluminescence of EVs in the liver, spleen, and lungs using the IVIS Spectrum system and demonstrated that EVs were taken up by macrophages in the liver and spleen and by the endothelial cells in the lungs [37].

Charoenviriyakul et al. used the same system (gLuc-lactadherin) in their studies to compare the pharmacokinetics and biodistribution of EVs obtained from five different cell lines, including murine melanoma, myoblast cells, fibroblasts, aortic endothelial cells, and macrophages. The visualization was performed using IVIS Spectrum 5 min after an intravenous injection. The authors demonstrated that EVs quickly disappeared from the bloodstream and were distributed mainly in the liver. The authors concluded that there is no difference in the distribution profile between EVs derived from different cell types [59].

Lai C.P. et al. have developed a sensitive and versatile reporter system for the visualization of EVs’ biodistribution. The system consists of Gaussia luciferase (Gluc) fused to the transmembrane domain of the platelet-derived growth factor receptor (PDGFR) and biotin acceptor domain (EVs-GlucB). The membrane-bound variant of the Gluc reporter serves to specify the signal from the EVs and biotin acceptor peptide serves to conjugate the reporter with streptavidin labeled magnetic nanoparticles, fluorophore, or radionuclide and expand the spectrum of detection methods. The authors isolated EVs from human embryonic kidney (HEK) 293T cells and injected them intravenously into mice. It was found that EVs first underwent a rapid distribution phase, followed by a longer excretion phase, mainly through the hepatic and renal pathways within six hours. The highest EV signals were detected 60 min after administration. The largest amount of bioluminescent signal was found in the spleen, liver, lungs, and kidneys [47].

Komuro H. et al. improved the specificity and enhanced EV-mediated cargo delivery to the pancreatic β-cell by modifying EVs’ surface with pancreas-specific p88 peptide. Lactadherin, which is known as a membrane-associated protein, was fused with p88 peptide and, as a result, cells and EVs contained p88 peptide on their surface. Additionally, donor 293T cells were transfected with plasmids encoding Gaussia luciferase (Gluc). It was demonstrated that signals from untargeted EVs were primarily found in the lungs and spleen, while bioluminescence signals from targeted EVs (containing p88 peptide) were predominantly observed in the lungs, spleen, and pancreas [60].

Gupta D. et al. created the tetraspanin (membrane proteins, enriched in EVs) fused NanoLuc or ThermoLuc fused proteins. The authors showed that the EVs’ distribution to various internal organs takes place within minutes after administration and EVs were mainly absorbed by the liver and spleen [50].

The limitations of the method include the labor-intensive procedure of establishing genetically modified producer cells of EVs, disturbed conformation of fused protein, and proteolytic cleavage of reporters associated with the EV surface [61]. Additionally, the BLI method is applicable mainly to small animals. To improve sensitivity of the method, additional hair removal manipulation or ex vivo analysis is required [62]. This method also requires the administration of luciferase substrates and is limited by low spatial and temporal resolution. For a more accurate analysis, the bioluminescence method is usually combined with other visualizing methods [63].

## 3. Imaging Using Vital Dyes

Organic vital dye have high brightness, excellent spatial resolution, and are used to visualize cells and EVs using microscopy or IVIS Spectrum. Currently, fluorescent vital dyes PKH2 and PKH26 [64,65,66], as well as dialkylcarbocyanine dyes such as DiD, Dil, DiO, and DiR [67], are widely used for EVs’ labelling (Table 1). Dyes having emission in the far-red region of the spectrum (such as PKH26 and DiD) are used for in vivo imaging, as most tissues do not autofluoresce in this range and these wavelengths have less phototoxicity [68,69].

The staining procedure consists of the direct incubation of either parental cells or the isolated EVs in the dye solution. Staining of parental cells with lipophilic dyes followed by isolation of labeled EVs makes it possible to thoroughly wash parental cells from unbound dye, but leads to a gradual loss of staining intensity of the cells and, accordingly, the released vesicles. EVs’ staining with lipophilic dyes makes it possible to obtain a high-intensity fluorescent signal; however, the washing step leads to the loss of EVs.

Grange C. et al. investigated the biodistribution of mesenchymal-stem-cell-derived EVs (MSC-EVs) using the model of acute renal failure. MSC-EVs were labeled with NIR dye directly or EVs were isolated from MSCs preincubated with the dye. The researchers found that EVs directly labelled with the dye showed higher and brighter fluorescence compared with EVs from labelled MSCs. MSC-EV accumulated in the kidneys of mice with acute renal failure, whereas this effect was not detected in control animals [69].

Wen S. et al. investigated the effect and biodistribution of MSC-EV in radiation-induced bone marrow damage in mice. They used lipophilic membrane DiD dye. It was observed that DiD-labelled MSC-EVs accumulated in liver and spleen and, to a lesser extent, in bone marrow, femur, and tibia, and was not found in the lungs, heart, and kidneys [70].

Mendt M. et al. studied the biodistribution of EVs derived from MSCs in a mouse model of pancreatic cancer. They used 1,1′-dioctadecyltetramethyl indotricarbocyanine iodide (DiIC18, XenoLight DiR) in their experiments. Labelled EVs were injected intravenously in mice. After 3, 6, 24, and 48 h, they were euthanized and internal organs (brain, kidney, spleen, liver, lungs, heart, pancreas, intestines, testes, and femur) were imaged immediately. It was found that EVs accumulated mainly in the pancreas and that the signal was higher than in the liver, spleen, and lungs. Preferential accumulation of EVs in tumor when EVs were injected intravenously in mice bearing tumors was also observed [71].

Wang D. et al. on the model of carotid artery damage in rats showed that MSC-derived EVs after intravenous administration distributed directly in the rat carotid arteries [72]. Wei Z. et al. studied the biodistribution and delivery efficiency of EVs derived from MSCs overexpressing the membrane protein CD47 on the model of acute myocardial infarction. The authors intravenously injected Dil-labeled EVs and found that CD47-EVs were detectable in plasma 120 min after injection, and some mice showed a Dil fluorescence signal even after 240 min [73]. EVs derived from stromal bone marrow cells were labeled with DIR and intravenously injected in 5T33MM mice. The results showed that EVs were distributed mainly in bone marrow, spleen, and liver [74]. 

Summarizing the described data, it can be concluded that the distribution profile of EVs depends on the route of administration and accompanying injuries and diseases. Systemic administration via tail vein demonstrates stable accumulation in the lungs, liver, and spleen, and less often in the pancreas and gastrointestinal tract.

Biodistribution of EVs depends on the cell source and route of administration. Wiklander O.P. et al. isolated EVs from C2C12 muscle cell lines, B16F10 melanoma cells, and primary immature dendritic cells (DCs) and stained them with lipophilic near-infrared dye DiR. The authors showed that the in vivo distribution of EVs from different cell sources was mostly similar. Liver, spleen, gastrointestinal tract, and lungs were the sites of the highest accumulation. The researchers also noted that EVs derived from C2C12 showed greater accumulation in the liver compared with B16F10-EV and EVs derived from DC. Conversely, accumulation in the lungs was lower for C2C12-EV. B16F10-EV was more frequently detected in the gastrointestinal tract, and DC-EV showed increased accumulation in the spleen [40]. Shi M.M. et al. labelled MSC-EVs with the fluorescent cell membrane dye DiR (1,1-dioctadecyl-3,3,3,3-tetramethylindotricarbocyanine iodide) and sprayed EVs using a nebulizer in mice. The study showed that the highest fluorescence intensity was observed in the lungs 24 h after spraying. Thereafter, a gradual decrease in intensity was observed by 28 days after administration [75].

Content and surface receptors of EVs influence their biodistribution. Nishida-Aoki N. et al. observed that deglycosylation of DiR-labelled EVs enhances their uptake by lung cells and, therefore, alter biodistribution in vivo [76]. 

A potential problem with the use of lipophilic dyes is the formation of micelles in fluid owing to their lipophilic nature [61]. It is a hard task to distinguish the signal of labelled EVs from the signal of a free dye, which can lead to inaccurate results of the biodistribution profile. Another problem is EVs’ phagocytosis by other cells and long-term unspecific persistence of a dye in vivo that can be misleading [77,78,79]. Therefore, labelling of EVs with a lipophilic dye is most suitable for short-term studies.

## 4. Tracking Using Fluorescent Proteins

Fluorescent proteins—green fluorescent protein (GFP), enhanced green fluorescence protein (EGFP), and red fluorescent protein (RFP)—are most commonly used to study biodistribution (Table 2). The procedure of EVs’ tracking using fluorescent proteins consists of obtaining a fused protein with an EV protein. For this purpose, the fluorescent protein gene is fused to the sequence of the gene encoding the EV marker protein (for example, CD9, CD63, and CD81). This gene is introduced into EV-producing cells using plasmid or viral vectors in order to further obtain EVs with fluorescence. The efficiency of the method and its resolution depends on the level of proteins’ expression, the labeling efficiency of EVs, and the excitation light source power [52].

To show the fate of EVs originating from cancer cells in orthotopic breast cancer models, Suetsugu A. et al. obtained EVs containing CD63-GFP and showed that cancer cells secreted GFP-labelled EVs into the tumor microenvironment [80]. In another study, EVs derived from breast cancer cells expressing CD63-RFP fusion protein on their surface were injected intravenously in mice. The authors showed that EVs promoted macrophage polarization, growth, and metastasis of the primary tumor [81].

Palmitoylation signal as a membrane-anchoring modification was used to trace EVs. The donor cells (EL4 thymoma cells) were labelled with EGFP fused with palmitoylation signal (PalmGFP) to study the communication between tumor cells and their microenvironment. The study showed that, 9 days after the subcutaneous implantation of tumor cells, EVs were visualized in a tumor. The amount of EVs was higher in peripheral areas of the tumor where the density of tumor-infiltrating cells is lower [82].

The described method of EVs’ tracking with a fluorescent protein has the disadvantage that the signal depends on the oxygenation level in the tissue. It was reported that decreased oxygenation leads to the disappearance of a fluorescence signal. Moreover, the modification of EVs’ surface proteins with a tag may affect their biodistribution [83].

According to the data described above, the biodistribution profile of EVs after intravenous injection was profiled based on the frequency of EVs’ detection in organs (Figure 1). EVs injected into the tail vein further entered the small circulatory system and reached the lungs, then EVs were distributed to the liver, spleen, and gastrointestinal tract (GI tract) and then detected in the kidneys and bladder (Figure 1). In general, a pattern of EVs’ accumulation in organs associated with the reticuloendothelial system was found. 

Small-EVs 15 min after intravenous administration could be detected in the bladder [84]. It is still unknown how EVs can penetrate the glomerular filtration barrier (GFB) and be found in urine. Under normal conditions, only <5–7 nm particles can be filtered through the glomerulus [85]. Nonetheless, EVs have been identified in the bladder [86] and urine [87]. 

## 5. Tracking Using Radioisotope Labeling (PET and SPECT)

Nuclear imaging is based on the administration and detection of decaying isotopes in vivo. When a radioisotope decays, positrons or gamma rays are emitted, producing two or one high energy photons. These photons are detected using positron emission tomography (PET) and single photon emission computed tomography (SPECT). Both PET and SPECT are used for molecular imaging thanks to their excellent sensitivity, contrast agent specificity, large field of view, good temporal resolution (from seconds to minutes), and taking less time [88].

PET is based on the detection of the radioisotope decay, which emits a positron and annihilates with an electron to form two high energy photons (511 keV emission). PET is based on the use of a contrast agent (indicator), which consists of a positron-emitting isotope (radioactive tag) bound to an organic ligand (target agent). The positron-emitting isotopes are ^11^C, ^13^N, ^15^O, ^18^F, ^64^Cu, ^68^Ga, ^76^Br, and ^94m^Tc [89,90]. The organic ligand interacts with the target protein, resulting in a characteristic distribution in the tissue. The ideal radiopharmaceutical for PET should only interact with the target protein and not give rise to accumulation phenomena [91]. The most common organic ligands with isotopes are as follows: fluorodeoxyglucose F-18 (FDG), acetate C-11, choline C-11, fludoxyglucose F-18, sodium fluoride F-18, fluoroethylspiperone F-18, methionine C-11, prostate-specific membrane antigen Ga-68 (PSMA), DOTATOC, DOTANOC, DOTATE Ga-68, florbetaben, florbetapyr F-18, rubidium chloride Rb-82, ammonia H-13, FDDNP F-18, oxygen-15 labelled water O-15, and FDOPA F-18 [92].

SPECT is similar to PET, implementing a contrast agent for imaging; however, this method uses heavier isotopes: ^133^Xe, ^99m^Tc, and ^123^I. These radioisotopes emit only one high-energy (gamma) photon upon decay and have longer attenuation times than those used in PET. SPECT detects gamma photons using a gamma camera, which rotates around the object in steps and generates an image [93].

EVs’ radiolabeling can be carried out through binding to the EVs’ surface or encapsulation [94]. For example, ^123^I and ^99m^Tc are able to form covalent bounds with EV surface functional groups. ^64^Cu, ^68^Ga, or ^111^In are also bound to the surface of EVs, but in the presence of bifunctional chelators (containing metal-binding moiety for radiometal sequestration). Lipophilic ^111^In-oxine or ^99m^Tc-HMPAO complexes cross membrane and accumulate in EVs [94]. Therefore, these isotope complexes are directly added into the suspension to label EVs.

Nuclear imaging has been used by many researchers to study EVs’ biodistribution (Table 3). For example, SPECT was applied by González M.I. et al. to investigate the biodistribution of EVs derived from goat milk. EVs were labelled with ^99m^Tc by mixing with ^99m^Tc solution at 37 °C and administered intravenously, intraperitoneally, and intranasally. Intravenously injected EVs rapidly disseminated and disappeared from the blood (<4 min) and were preferentially absorbed in the liver and spleen. After intraperitoneal administration, EVs circulated in the blood stream three to four times longer compared with intravenous injection and were uptaken mostly by pulmonary tissue. Intranasal administration showed EVs’ accumulation in the nose and no brain tissue penetration was detected [95].

Varga Z. et al. and Gangadaran P. et al. used SPECT to trace EVs derived from red blood cells. EVs were labelled with ^99m^Tc-tricarbonyl complex by incubating them in a labelling solution for 30 min at room temperature. Gangadaran P. et al. labelled EVs with ^99m^Tc in the presence of 0.01% stannous chloride with subsequent purification of EVs using columns. The authors found that intravenous administration of labelled EVs resulted in their accumulation in the liver and spleen [44,96]. EVs derived from macrophages also accumulated in the liver and spleen and, 3 h after injection, were found in salivary glands, and no brain tissue penetration was detected [41]. Nagelkerke A. et al. characterized the pharmacokinetics and distribution of ^125^I labelled EVs derived from Raw 264.7 cells in CD-1 mice following intravenous injection. The study showed that EVs after 10 min, 4 h, and 24 h accumulated mainly in the liver and spleen. Accumulation in the brain was negligible and after 10 min and, after 4 h, it was 0.093% [84].

Biodistribution of EVs derived from tumor cells was investigated using SPECT technique. Smyth T. et al. labelled EVs from PC3 and MCF-7 cells with Indium-111 and injected them intravenously into athymic mice with PC3 tumors and mice without tumors. EVs were labelled with 111 In-oxine by incubation in a staining solution for 20 min at room temperature with subsequent purification using columns. Analysis showed rapid elimination of both PC3 and MCF-7 EVs from the blood. Three hours after injection, less than 5% of the injected dose remained in the bloodstream. Twenty-four hours after injection, most of the EVs were distributed in the liver, spleen, and kidneys. Researchers also found that the presence of a tumor had no significant effect on EVs’ biodistribution [39]. These results were confirmed in the study of Faruqu F.N. et al. EVs derived from B16F10 cells were labelled with Indium-111 and injected i.v. The authors applied the approaches of intraluminal and membrane labelling. For intraluminal labeling, the Indium-111-tropolone complex was added directly to EVs. For membrane labelling, EVs were incubated with Indium-111 in the presence of DTPA anhydride. After the injection, EVs were visualized in the liver, spleen, and urinary bladder for 30 min and no accumulation of EVs in the tumor was detected [42]. In contrast, Rashid M.H. et al., using Iodine-131 labelled EVs derived from 4T1 and AT3 cells, showed that, 3 h after i.v. injection, a sufficiently strong radioactive signal was detected at the primary tumor site and the site of metastasis (lung) [97]. This can be explained by the origin of the injected EVs from different cell types and the possible influence of the type of radioactive label on the efficiency of biodistribution analysis. 

Molavipordanjani S. et al., using SPECT, found that most of the radioactivity was found in the liver and kidneys, while other organs, including the stomach, bones, and thyroid, showed rather low levels of radioactivity. The authors used fac-[^99m^Tc(CO)_3_(H_2_O)_3_]^+^ complex for direct labelling of EVs. The uptake of radioactivity by tumor tissue 1 and 4 h after i.v. injection was 2.75 and 1.47% ID/g (percentage of injected dose of radioactivity per gram of tissue), respectively [98]. Direct injection of ^125^I labelled EVs derived from B16BL6 cells into a tumor led to the following distribution: after 1 h—57%, after 4 h—43%, after 8 h—33%, and after 24 h—34% of signal remained in the tumor. Distribution of the signal to other organs was fairly limited up to 48 h after injection [99]. The same EV type (derived from B16BL6 cells) and tracking system was used in the study of Morishita M. et al. The authors showed that intravenously administered EVs quickly disappeared from the bloodstream. After 4 h, they were detected in the liver, spleen, and lungs [100].

Royo F. et al. using PET found that intravenous injection of EVs derived from proliferative mouse liver cells leads to a rapid accumulation of EVs in the liver. A few hours later, EVs were also distributed in other organs, including the brain. Glycosidase treatment caused EVs’ accumulation in the lungs [101].

One of the disadvantages of the nuclear imaging is the low spatial resolution of the PET and SPECT techniques, linked to difficulties in pinpointing the source of radiation in the tissue [89]. Another disadvantage is the use of hazardous radioisotopes [102]. Some of them are sources of rather high doses of radiation, which are dangerous for both research subjects and researchers themselves [43,43,63]. Additionally, the high cost of reporters and requirement of specialized equipment make this method available on a limited basis to researchers [103].

## 6. Tracking Using CT and MRI

Tomography is an effective method of non-invasive EV tracking. Tomographic imaging has excellent penetrating power and high spatial and temporal resolution. To perform tomography, EVs are tagged with nanoparticles. Commonly used imaging equipment for tomographic imaging includes computed tomography (CT) and magnetic resonance imaging (MRI) [104].

Hu L. et al. used MRI to investigate the biodistribution of EVs derived from mouse melanoma B16-F10 cells (CRL 6475). They performed electroporation to load superparamagnetic iron oxide nanoparticles (SPIO) in EVs. The study demonstrated that labelling had no effect on EVs’ size and biodistribution in lymph nodes [104]. Iron oxide nanoparticles were used by Busato A. et al. to load into EVs derived from stem cells. Visualization of labelled EVs in mice was performed after intramuscular injection. The study showed that EVs were clearly detected in muscle tissue after injection [105]. 

SPIO were also used by Jung K.O. et al. to visualize EVs derived from MDA-MB-231 human breast cancer cells in vivo using MRI after i.v. injection. For SPIO labeling of EVs, parental cells were incubated with SPIO nanoparticles for 24 h. One hour after injection, uptake of EVs in the brain, heart, lungs, liver, spleen, kidney, and intestine was observed. In vivo and ex vivo studies showed that EVs accumulated predominantly in the liver [106]. 

Bose R.JC et al. used CT and MRI techniques in their study. They labelled tumor cell-derived EVs with gold-iron oxide (GION) nanoparticles and injected them intravenously in mice bearing 4T1 tumors. For labeling of EVs with gold-iron oxide (GION) nanoparticles, the authors performed their co-extrusion through a 200 nm pore sized membrane. Visualization was performed 12 days after three intravenous injections (days 1, 4, and 8). These studies showed that strong signals of labelled EVs were detected in the tumors [107].

Betzer O. et al. used CT and glucose-coated gold nanoparticles to track EVs after intranasal and intravenous administration on the model of focal cerebral ischemia in vivo. The authors labeled EVs with glucose-coated gold particles by direct incubation, as glucose-coating facilitates the uptake of particles by EVs. The experiments showed an increased accumulation of EVs at the lesion site within 24 h after intranasal administration, whereas intravenous administration led to nonspecific distribution [108].

A disadvantage of MRI is the difficulty in obtaining and quantifying whole-body images [43].

## 7. Novel Tracking Approaches

The photoacoustic imaging or tomography (PAI) method combines the photoacoustic effect with ultrasound imaging. Photoacoustic imaging is a new hybrid imaging technique that can provide strong contrasts of endogenous and exogenous optical absorption with high spatial ultrasound resolution. PAI is a non-invasive pulsed laser light that is absorbed by contrast agents and converted into acoustic signals. PAI has deep tissue penetration and high spatial resolution [109].

This method was used by Ding H. et al. for tracking of exosome-like vesicles. The authors used the oxidized form of 2,2′-azino-bis (3-ethylbenzothiazoline-6-sulfonic acid) (ABTS) as a contrast photosensitive agent, which has strong near-infrared absorption (absorption maxima at 734 nm). In vivo experiments have shown that these vesicles accumulate effectively in a tumor [110]. Piao Y.J. et al. applied this method to study the effect of exosomes derived from triple negative breast cancer (TNBC) cells on metastatic processes in lymph nodes [81]. Cao T.G.N. et al. used indocyanine green (ICG) as a contrast agent, which was loaded into the EV along with paclitaxel (PTX) and sodium bicarbonate by incubation. PAI signals were measured at 780 nm [111]. Lv S. et al. used MRI and PAI to investigate the biodistribution of tumor-derived EVs. Water-soluble gadolinium-based melanin nanoparticles (MNP-Gd) were used as a contrast agent. Labeling of EVs with a contrast agent was carried out by incubation overnight at 4 °C. The study showed that, after intravenous injection, EVs accumulated in tumors and liver were then metabolized by the liver and kidneys [112].

Another relatively new bioimaging technique is Raman spectroscopy, which is based on inelastic scattering. This is a fairly highly sensitive tool for molecular imaging. A beam of light with a certain wavelength is passed through the studied sample; upon contact with the sample, it is scattered. The scattered rays are collected into a single beam by a lens and passed through a light filter, which separates the weak (0.001% intensity) Raman rays from the more intense ones. The ‘pure’ Raman rays are amplified and directed to a detector, which records their frequency [113,114]. This method provides information about the biochemical components of the sample. Horgan C.C. et al. labelled isolated EVs with deuterium, which acts as an active tag for Raman spectroscopy. The authors showed that this method provides both spectral analysis of EVs’ composition and high-resolution spectroscopic imaging of EVs in vitro [115]. However, one of the main limitations of Raman spectroscopy is low sensitivity, as only 1 of 10^6^ photons are Raman scattered. It was observed that the signal can be enhanced using a substrate made of a noble metal (usually silver or gold) by up to 10^11^ [116]. Therefore, this method was called surface-enhanced Raman spectroscopy (SERS).

Ćulum N.M. et al. used SERS for the analysis of EVs. The authors made gold nanohole arrays of varying sizes and shapes for trapping single EVs and enhancing their vibrational signature [117]. Zhang H. et al. used Raman spectroscopy to differentiate EVs from bovine placenta (trophoblast) and peripheral blood mononuclear cells [118]. Raman spectroscopy was used to detect biochemical differences in EVs isolated from plasma of patients with sporadic amyotrophic lateral sclerosis [119].

Raman spectroscopy has also been used to characterize different EVs’ fractions in patients with prostate cancer [120]. Samoylenko A. et al. applied SERS to analyze the EVs of renal carcinoma [121]. Chalapathi D. also used SERS and citrate-silver nanoparticles to distinguish EVs derived from cancerous cells based on their difference in chemical compositions [122].

## 8. Conclusions

Methods of labeling and tracking EVs to detect their function and behavior in vivo have reached a sophisticated level. We summarized in this review the methods known to date and discussed their advantages and limitations to facilitate the selection of the most suitable method for the purpose of the research. It was established that radioisotopes and lipophilic vital dyes are the most sensitive tracers for in vivo imaging of EVs. Radioactivity is additionally the most accurate approach, whereas, fused with surface proteins, luciferase molecules can alter EVs’ distribution, resulting in high accumulation in the lungs [123]. All of the above-mentioned detection methods found the same pattern of EVs’ biodistribution. After intravenous injection, the main organs in which EVs, regardless of origin, accumulated were the liver, lungs, spleen, and kidneys. Most of the EVs accumulated in the liver, whereas in the lungs, spleen, and kidneys, the amount of EVs was 5–6 times lower. After intraperitoneal administration, EVs localized mainly in the liver, lungs, spleen, gastrointestinal tract, and pancreas. After intranasal administration, EVs were detected in the lungs and brain.

It seems that EVs’ half-life depends on the cell source. EVs of most cell types have a half-life in blood of about 10–30 min [47,123]. However, human-platelet-derived EVs remained in the circulation with a half-life of 5.5 h [124]. For the rapid clearance of EVs from the blood, macrophages related to the reticuloendothelial system are primarily responsible. Macrophages recognize the negative charge of phosphatidylserine (PS) enriched in EVs and capture them [125]. Recently, in a non-human primate model, it was shown that EVs following intravenous administration were uptaken by PBMCs, most notably B-cells, but remained detectable in plasma for up to 24 h. Then, within one hour, strong signals from EVs in the liver and spleen and some uptake in the lungs were detected [126]. 

The described methods of in vivo tracking have their advantages and limitations. Therefore, dual labeling of EVs or few detection methods are used to complement the aforementioned limitations and arrive at accurate conclusions. For example, SPECT is used in combination with CT to obtain more complete anatomical and functional information. Fluorescence staining, bioluminescence, and radioisotopes were applied to enable accurate spatiotemporal resolution of EV trafficking [123].

The development of new reporters and techniques for EVs’ imaging is an actual field of research that will help to obtain a more accurate profile of EVs’ distribution, improve our understanding of EVs’ participation in physiological and pathological processes, and develop a clinically approved therapeutic tool based on EVs.

## Figures and Tables

**Figure 1 ijms-23-11312-f001:**
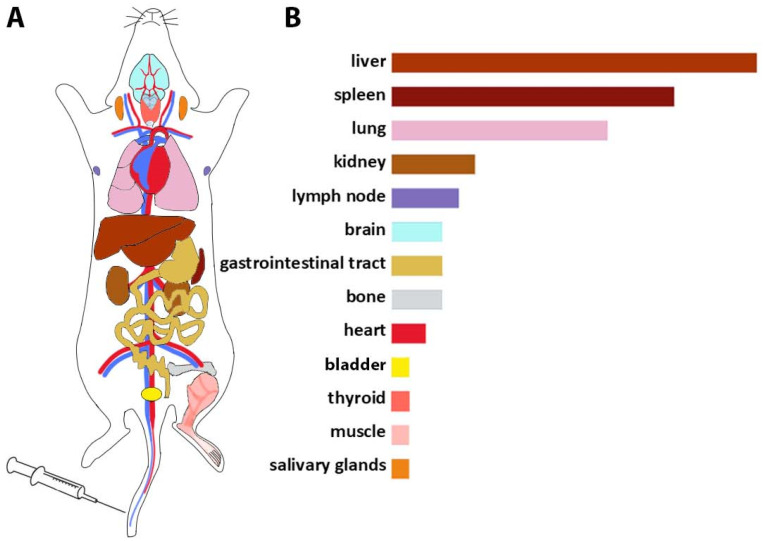
Biodistribution profile of EVs after i.v. injection into the tail vein. (**A**) Route of EVs’ distribution; through the bloodstream, EVs enter the small circle of the circulation, enter the lungs, then accumulate in the liver and spleen, are detected in the GI tract, and finally are detected in the kidneys and bladder. (**B**) The frequency of detection of EVs after i.v. administration in organs (based on the reviewed data).

**Table 1 ijms-23-11312-t001:** The most commonly used bioluminescent tags used to track EVs.

№	EVs Donor Cells	Specimen	Reporter	Administration	Biodistribution Profile	Reference
1	Thyroid cancer cells (CAL-62) and breast cancer (MDA-MB-231)	Mice	Luciferase from Renilla reniformis (Rluc)	i.v., s.c.	Liver, spleen, kidneys, lungs	[56]
2	Human embryonic kidney cells (HEK) 293T	Mice	Reporter PalmGRET; Gluc, fused to the acceptor domain of biotin (GlucB); NanoLuc, ThermoLuc	i.v.	Liver, spleen, lungs, some kidneys	[58] [47] [50]
3	Mouse melanoma cells B16-BL6	Mice	Luciferase gaussia and truncated lactadherin (gLuc-lactaderin)	i.v.	Liver, lungs, spleen	[48] [37]
4	Mouse melanoma cells, myoblast cells, fibroblasts, aortic endothelial cells, macrophages	Mice	gLuc-LA	i.v.	Liver	[59]
5	Mouse breast cancer cells (MET-1)	Mice	Luciferase (Luc)	i.v.	Pancreas, abdominal wall	[57]
6	293T cell line	Mice	Gaussia-luciferase (Gluc)	i.v.	Lungs, spleen, pancreas	[60]

i.v.—intravenous, s.c.—subcutaneous.

**Table 2 ijms-23-11312-t002:** Most commonly used lipophilic dyes and fluorescent proteins.

№	EV Sources	Specimen	Reporter	Administration	Biodistribution Profile	Reference
1	Mesenchymal stem cells	Mice with acute renal failure	DiD	i.v.	Kidneys (for acute renal failure)	[69]
2	Mesenchymal stem cells	Mice	DiD	i.v.	Liver and spleen and, to a lesser extent, spinal marrow, femur, tibia	[70]
3	Mesenchymal stem cells	Mice	DiR	i.v., i.p.	Pancreas, liver, spleen, lungs, heart, tumor	[71]
4	Mesenchymal stem cells	Rats	DIO	i.v.	Carotid arteries of rats	[72]
5	5T33 Mouse stromal stem cells	5T33MM Mice	DIR	i.v.	Liver, spleen	[74]
6	C2C12, B16F10, dendritic cells (DC)	Mice	DiR	i.p., s.c., i.v.	Liver, spleen, gastrointestinal tract and lungs	[40]
7	Breast cancer cells (MCF10A)	Mice	DiR	i.v.	Liver, spleen, some brain	[76]
8	Overexpressed CD47 mesenchymal stem cells	Mice	Dil	i.v.	Liver, spleen, pancreas	[73]
9	Mesenchymal stem cells from adipose tissue	BALB/c Mice	DiR	Intratracheal	Lungs	[75]
10	MMT-060562 mice breast cancer cells (MMT) and MDA-MB-231 human breast cancer cells	Nude mice	CD63 with green fluorescent protein (GFP)	In the adipose tissue of the breast	Lungs	[80]
11	MDA-MB-231 cancer cells line	Mice	RFP, DiD	i.v.	Axillary lymph nodes	[81]
12	Implanted thymoma EL4	C57BL/6 Mice	EGFP	EV in tumor, implanted thymoma	Tumor and surrounding tissue	[82]

i.v.—intravenous, i.p.—intraperitoneal, i.n.—intranasal, i.m.—intramuscular injection, s.c.—subcutaneous.

**Table 3 ijms-23-11312-t003:** Most commonly used nuclear imaging indicators and/or isotopes in EVs’ biodistribution study.

№	EVs’ Source	Specimen	Tag	Administration	Biodistribution Profile	Reference
1	Goat milk	Balb/C mice	Technetium-m	i.v., i.p., i.n.	Liver, spleen, lung tissue	[95]
2	PC3, MCF-7	Mice	Indium-111	i.v.	Liver, spleen, kidneys	[39]
3	Proliferative mouse liver cells	Mice	[124 I] Na	i.v., into the hock	Liver, brain, lung tissue, axillary lymph nodes	[101]
4	B16F10 cell culture	C57Bl/6 mice	Indium-111	i.v.	Liver, spleen, bladder	[42]
5	B16BL6 mice melanoma cells	Mice	Iodine-125	i.v.	Liver, spleen, lungs	[100]
6	B16BL6 mice melanoma cells	C57BL/6J mice	Iodine-125	Into the tumor	Tumor	[99]
7	4T1 и AT3 cell lines	Mice	Iodine-131 (131 I)	i.v.	Tumor, lungs	[97]
8	Red blood cells	Mice	Technetium-m tricarbonyl complex	i.v.	Liver, spleen	[96]
9	HEK293T with a chimeric gene LAMP2b-DARPin G3	SKOV-3 and BALB/c mice with tumors	Technetium-m	i.v.	Liver, intestines, kidneys, bones, thyroid gland	[98]
10	RAW264.7Macrophages	Mice	Technetium-m	i.v.	Salivary glands, liver, spleen, brain, intestines	[41]
11	Red blood cells	Mice	Technetium-m	i.v.	Liver, spleen	[44]
12	Raw 264.7 cell culture	Mice	Iodine-125	i.v.	Liver, spleen	[84]

i.v.—intravenous, i.p.—intraperitoneal, i.n.—intranasal, i.m.—intramuscular injection, s.c.—subcutaneous.

## Data Availability

All data generated or analyzed during this study are included in this published article. The data that support the findings of this study are available from the corresponding author upon request.

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
