# Peer review of "Tracking of Extracellular Vesicles’ Biodistribution: New Methods and Approaches"

_ijms, 2022, doi:10.3390/ijms231911312_

Round 1

Reviewer 1 Report

This paper reviewed recent in vivo EV labelling and tracking technologies, which include engineering EVs with reporter proteins for bioluminescence imaging, labelling EVs with fluorescent vital dye or fluorescent proteins for imaging, labelling EVs with radioisotopes for nuclear imaging, and nanoparticle labelling for tomography imaging. The authors also discussed both advantages and disadvantages of each method, and compared EV distribution profiles.

Below are some suggestions.

1. It will be good to include a bit more details about each labelling method as it is an important part of EV biodistribution. For example, the methods for EV radioisotope labelling or nanoparticle labelling (sonication, ester conjugation…), the methods for fluorescent protein labelling (labelling cells, labelling EVs…)

2. There are a bit more can be included in the discussion of advantage and disadvantage of each method. For example, with dialkylcarbocyanine dyes, it is hard to separate the signal of labelled EVs from the signal of dyes only, thus the biodistribution profile may not be accurate.

3. The conclusion of EV biodistribution profile was about different administration routes. It will be good if include a conclusion of EV biodistribution file based on method comparison.

Author Response

Reviewer 1

Comment:  It will be good to include a bit more details about each labelling method as it is an important part of EV biodistribution. For example, the methods for EV radioisotope labelling or nanoparticle labelling (sonication, ester conjugation…), the methods for fluorescent protein labelling (labelling cells, labelling EVs…).

Reply: Thank you for the suggestion. We have added information about the labeling of extracellular vesicles (Lines 190-195, 253-257, 324-330, 333, 342-345, 356-357, 363-366, 376-377, 416-417, 423-424, 429-430).

Comment:  There are a bit more can be included in the discussion of advantage and disadvantage of each method. For example, with dialkylcarbocyanine dyes, it is hard to separate the signal of labelled EVs from the signal of dyes only, thus the biodistribution profile may not be accurate.

Reply: As it was suggested, we have added information about the advantages and disadvantages of the methods (Lines 172-174, 245-247, 278-279, 399-400).

Комментарий : Заключение профиля биораспределения EV было связано с различными путями введения. Было бы хорошо, если бы вы включили заключение файла биораспределения EV на основе сравнения методов.

Ответ: Как и было предложено, мы сравнили профиль биораспределения ЭВ, полученный разными методами, и обнаружили, что он практически одинаков, за исключением данных, полученных с помощью системы детектирования люциферазы. Мы обсуждали это в строках 487-496.

Reviewer 2 Report

1) Please provide the reported values of SNR with bioluminescence imaging, in general; and particularly in the context of EVs.

2) Some discussion should be provided on the reported range of the EVs’ half-life in blood, discussion of the factors that affect half-life of EVs in blood, and whether cells from which they are derived is also a factor or not.

3) I suggest adding a Table to summarize the key information provided in section 2 (bioluminescence).

4) Many of the EVs types mentions are too large (> 10 nm) to be excreted through the kidneys as the authors state (Figure 1). Were they detected in urine?  Revisions are necessary.

5) It should be stated what the molecular source of contrast is in photoacoustic imaging as related to EVs. In other words, what is the intrinsic molecule(s) that absorbs the light to give rise to the signal?  Also the wavelengths used in photoacoustic imaging associated with EVs?

6) Possible to provide some quantification of the relative fraction of EVs in the various organs. Also information on quantitative pharmacokinetics seems to be missing.

7) It should be clarified that Raman scattering relies on inelastic scattering. Given the weak signals associated with this method, if there are any limitations, they should be pointed out.

There are new methods based on surface-enhanced Raman spectroscopy that should also be pointed out.

8) The manuscript will benefit by adding some additional information that summarizes how these various imaging methods complement each other, and whether some common findings/result  emerge from the different methods.

9) Overall, section 7 is somewhat superficial and should be enhanced/elaborated.

Author Response

Reviewer 2

Comment:  Please provide the reported values of SNR with bioluminescence imaging, in general; and particularly in the context of EVs.

 Reply: Thank you for the suggestion. We have added this information in Lines 82-86.

Comment:  Some discussion should be provided on the reported range of the EVs’ half-life in blood, discussion of the factors that affect half-life of EVs in blood, and whether cells from which they are derived is also a factor or not.

 Reply: As it was suggested we have added information about the half-life of EVs from different cell types in blood and discussed the factor that affect half-life of EVs (Lines 497-505).

Comment:  I suggest adding a Table to summarize the key information provided in section 2 (bioluminescence).

 Reply: We have added the suggested table (from Line 170).

Comment:  Many of the EVs types mentions are too large (> 10 nm) to be excreted through the kidneys as the authors state (Figure 1). Were they detected in urine?  Revisions are necessary.

 Reply: We have edited the caption of Figure 1 to be more accurate in description findings (Line 292). And we have added information about the detection of EVs in kidneys, bladder and urine (Lines 295-298).

Comment:  It should be stated what the molecular source of contrast is in photoacoustic imaging as related to EVs. In other words, what is the intrinsic molecule(s) that absorbs the light to give rise to the signal?  Also the wavelengths used in photoacoustic imaging associated with EVs?

 Reply: As it was suggested we have added the required information (Lines 443-446, 449-456).

Comment:  Possible to provide some quantification of the relative fraction of EVs in the various organs. Also information on quantitative pharmacokinetics seems to be missing.

 Reply: As it was suggested we have added a distribution profile (Lines 493-506).

Comment:  It should be clarified that Raman scattering relies on inelastic scattering. Given the weak signals associated with this method, if there are any limitations, they should be pointed out.

There are new methods based on surface-enhanced Raman spectroscopy that should also be pointed out.

 Reply: Thank you for pointing this out. We have added the missing information (Line 457), discussed the limitation of Raman spectroscopy (Lines 467-471) and expanded the overview of this method (highlighted in blue, Lines 449-456, 472-482).

Comment:  The manuscript will benefit by adding some additional information that summarizes how these various imaging methods complement each other, and whether some common findings/result  emerge from the different methods.

 Reply: As it was suggested we have added additional information about complementation of the described methods (Lines 507-512). As well we have discussed the biodistribution profile of EVs obtained by different methods (Lines 492-497).

Comment:  Overall, section 7 is somewhat superficial and should be enhanced/elaborated.

 Reply: As it was suggested we have enhanced and expanded this Section (highlighted in blue, Lines 443-446, 449-456, 467-482).

Round 2

Reviewer 2 Report

Authors have addressed my previous comments adequately. Thank you. Table numbers need to be corrected.